

# Applications, challenges, and solutions of unmanned aerial vehicles in smart city using blockchain

Syed Faisal Abbas Shah[1,*], Tehseen Mazhar[1,*], Tamara Al Shloul[2], Tariq Shahzad[3], Yu-Chen Hu[4], Fatma Mallek[5] and Habib Hamam[5,6,7,8]

[1] Department of Computer Science & Information Technology, Virtual University of Pakistan, Lahore, Pakistan
[2] Department of General Education, Liwa College of Technology, Abu Dhabi, United Arab Emirates
[3] School of Electrical Engineering, Department of Electrical and Electronic Engineering Science, University of Johannesburg, Johannesburg, South Africa
[4] Department of Computer Science & Information Management, Providence University, Taichung City, Taiwan
[5] Faculty of Engineering, University of Moncton, Moncton, Canada
[6] College of Computer Science and Engineering, University of Ha'il, Ha'il, Saudi Arabia
[7] International Institute of Technology and Management, Libreville, Commune d'Akanda, Gabon
[8] Spectrum of Knowledge Production & Skills Development, Sfax, Tunisia
* These authors contributed equally to this work.

Corresponding authors
Syed Faisal Abbas Shah,
syedfaisalshah196@gmail.com
Tehseen Mazhar,
tehseenmazhar719@gmail.com

## ABSTRACT

Real-time data gathering, analysis, and reaction are made possible by this information and communication technology system. Data storage is also made possible by it. This is a good move since it enhances the administration and operation services essential to any city's efficient operation. The idea behind "smart cities" is that information and communication technology (ICTs) need to be included in a city's routine activities in order to gather, analyze, and store enormous amounts of data in real-time. This is helpful since it makes managing and governing urban areas easier. The "drone" or "uncrewed aerial vehicle" (UAV), which can carry out activities that ordinarily call for a human driver, serves as an example of this. UAVs could be used to integrate geospatial data, manage traffic, keep an eye on objects, and help in an emergency as part of a smart urban fabric. This study looks at the benefits and drawbacks of deploying UAVs in the conception, development, and management of smart cities. This article describes the importance and advantages of deploying UAVs in designing, developing, and maintaining in smart cities. This article overviews UAV uses types, applications, and challenges. Furthermore, we presented blockchain approaches for addressing the given problems for UAVs in smart research topics and recommendations for improving the security and privacy of UAVs in smart cities. Furthermore, we presented Blockchain approaches for addressing the given problems for UAVs in smart cities. Researcher and graduate students are audience of our article.

## INTRODUCTION

Information technology (IT), robotics, and artificial intelligence are just a few innovative services and technologies smart cities use to promote creativity. Uncrewed aerial vehicles (UAVs) are one of the main fields of research and development. UAVs are now more practical, dependable, secure, and able to operate for extended periods due to advancements made in this area for military and civilian purposes. Technology advancement is one of the most exciting and productive evolution. UAVs can be utilized inside and outside smart cities for various purposes, including agriculture, traffic management, monitoring, mapping, emergency services, weather monitoring, resource discovery, and environmental analysis.

UAVs have recently emerged as one of the most important and quickly growing topics of interest for commercial, military, and personal purposes (*Jain et al., 2021*). These technologies will contribute to developing intelligent and automated services, increasing infrastructure functionality, and the comfort resident's experience. The delivery of cost-effective infrastructures and services is the primary objective of a smart city design, which aims to achieve this goal (*Hameed, 2019*). The increase in UAV/drone usage is attributed to various factors, including increased demand for live streaming, real-time video filming, picture capture capabilities, mobility, and ease of use. UAVs are one of the many technologies that have the potential to make significant contributions in this area, and they are one of the technologies that could become an integral part of smart cities. UAVs are valuable for security, monitoring, and rescue operations (*Mohamed et al., 2020*). UAVs are mobile platforms that are both flexible and quick and that can be employed in the development of smart cities. Due to their easy deployment, easy operation, unique hovering capability, and excellent mobility, UAVs have also been used for small goods distribution and delivery (*Yaacoub et al., 2020*). UAVs are more adaptable and can operate in various places and situations humans find difficult. They can also fly closest to targets, allowing for more precise measurements and targeted operations (*Alfeo, Cimino & Vaglini, 2019*). Figure 1 shows the architecture of the UAV system.

These characteristics are beneficial when UAVs are employed for smart city applications. There are many challenges to UAVs in smart cities. The regulations for using UAVs in urban areas are still evolving, and there are many challenges in terms of policy, privacy, and safety (*Alfeo, Cimino & Vaglini, 2019*). UAVs face many technical challenges, such as limited flight time, payload capacity, and communication limitations in urban environments. UAVs must be integrated with other systems, such as traffic management, security, and emergency response systems. There will be certain challenges in implementing UAVs in smart cities. Large volumes of private information are generated in UAVs. Protecting individuals' privacy while ensuring that this data can be received, stored, and processed securely is a major difficulty. This article examines the various challenges associated with UAV deployment in urban areas and the ways in which blockchain technology could assist to overcome them. Blockchain is a decentralized digital ledger that secures and records transactions across a network of computers using encryption technology. Data from UAVs and other Internet of Things (IoT) devices can be safely

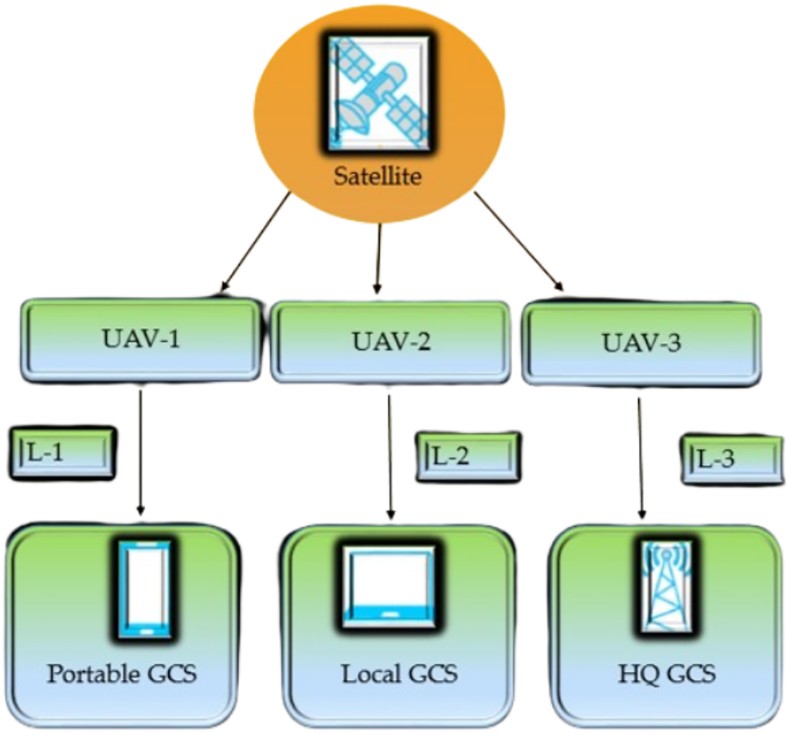

**Figure 1 Architecture of UAV system.**

exchanged and checked over a blockchain-based network, strengthening the security fabric of smart cities (*Alvares, Silva & Magaia, 2021*). When combined with drones, Blockchain technology has the potential to provide new solutions in a variety of industries, including supply chain management, delivery services, and safe data transmission. An overview of a smart city with drones is captured in Fig. 2.

This research contributes to finding the challenges associated with current and future UAV applications in smart cities. In this article, we have conducted the research to ensure that it is appropriate to the objectives of our article. At the next level, the related work is described. In addition, the UAV types and their applications are discussed in a smart city. The next topic is about the challenges which are serious challenges of UAVs in the smart city environment. The articles that did not concentrate on UAVs, applications, and blockchain were excluded. UAVs are mobile platforms used in today's smart cities for various tasks. These technologies could be used to keep updated on issues like population density, traffic flow, the environment, and public safety, among other things. UAVs can be successfully used in cities by developing applications suited to the needs and capabilities of smart cities. Uncrewed aerial vehicles (UAVs) can be used in smart cities to check and manage infrastructure because they are built with monitoring sensors, cameras, and modern software. There are various advantages to UAVs in smart city applications. UAVs are more adaptable and less expensive to operate. Therefore, they can carry out tasks in more places and under more demanding conditions than people. They may also go

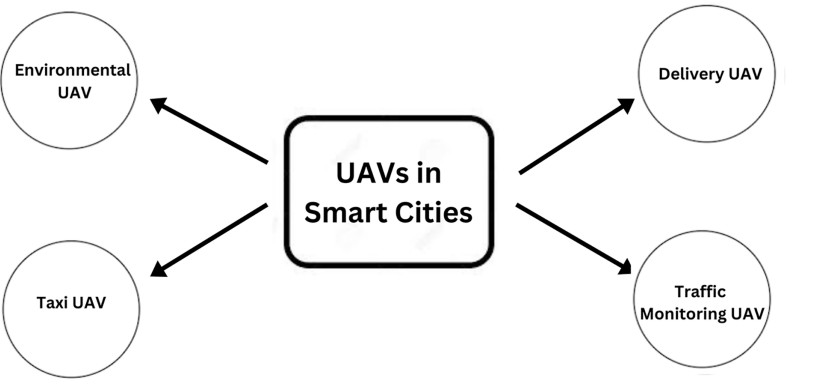

**Figure 2** Drones in smart cities.     

significantly closer to their targets, allowing for more accurate measurement and weapon deployment.

The rest of the article is structured as follows. A brief literature review analysis to help readers better comprehend the historical context of UAVs and blockchain technology is discussed in "Related Work". In "Methodology", the methodology and research questions are written with detail. In "What are the Applications and Types of UAVs?", the different types and applications of UAVs are described. In "What are the Challenges in UAVs in Smart Cities?", vulnerabilities and challenges to UAVs in smart cities are discussed. The solution and to challenges with the integration of blockchain with UAV are described in "What are the Solutions to Key Challenges with the of Integration Blockchain with UAVs?". "Conclusion and Future Direction" deals with the conclusion and future directions.

## RELATED WORK

UAVs were initially developed for defense purposes. However, research has revealed tremendous potential in various domestic applications. Problems with insufficient energy and computing resources are a common challenge for drones. Researchers have provided new insights that can help improve existing solutions to these problems (*Naqvi et al., 2018*). In *Fotouhi et al. (2019)*, the authors presented findings from a study on how UAVs helped improve the cyber-physical protection of cellular communications. In *Yan et al. (2019)*, the authors investigated, examined, and simulated air-to-ground channel measures for UAV communication. They studied the link quality for UAV communications and offered a design guideline for link budget control. Applications designed for smart cities can benefit from the varied computing, storage, and advanced service capabilities provided by cloud computing. According to *Mohamed, Lazarova-Molnar & Al-Jaroodi (2017)*, Data analysis, deep learning, simulations, optimization, and autonomous decisions are some examples of advanced services. In *Gupta, Jain & Vaszkun (2015)* discussed research on UAV communications in routing, smooth turnover, and power efficiency.

An overview of the literature concerning computer ethics and safety was presented by *Bagloee et al. (2016)*, which covered topics such as the connectivity of vehicles and the infrastructure necessary for the future expansion of autonomous vehicles (AVs).

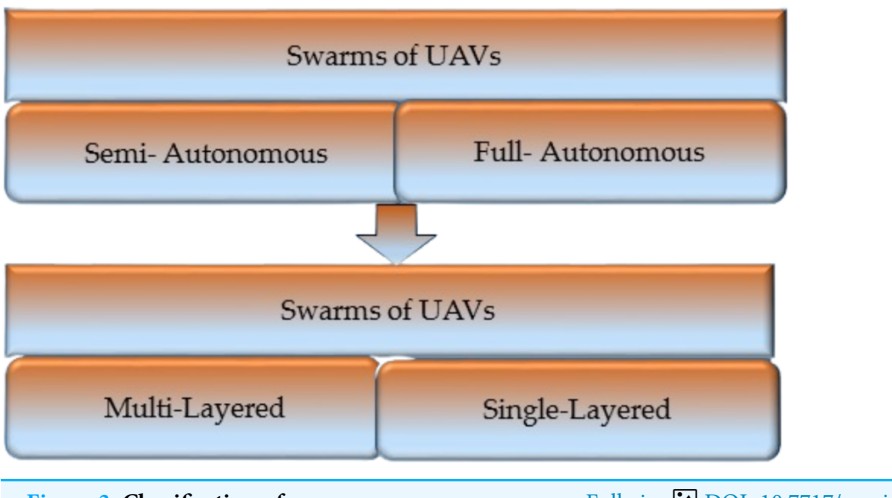

**Figure 3 Classification of swarms.**

According to the study, there is a substantial knowledge gap in UAV technology regarding the activities that occur across routes. To ensure the reliability of UAV communication systems, several researchers have included artificial intelligence, navigational techniques, cryptography, and low-latency communication systems among nodes of the UAV network (*Zhao et al., 2018*). The author *Rejeb et al. (2021)* investigated the capabilities of drones to learn about their achievements and difficulties in the field of logistics for humanitarian applications. They comprehensively assessed humanitarian drones' capabilities, problems, and results in logistical operations, administration, and governance. *Raciti, Rizzo & Susinni (2018)* presented a strategy devised for landing UAVs and charging their batteries on the top of a building. The authors of this study have also addressed misalignment losses and potential strategies to overcome them. In *Noor, Abdullah & Hashim (2018)*, the authors discuss the growth of UAV/drone remote sensing applications in urban environments and how these applications might resolve issues correspondingly. The author also discovered that developing UAV and drone-based remote sensing effectively resolves contemporary urban issues while ensuring urban areas' resilience and long-term viability. Figure 3 (*Tahir et al., 2019*) shows the classification of swarms.

The article's authors *Galkin, Kibilda & DaSilva (2017)* model the probability that a user will have a line-of-sight (LOS) channel to a UAV as a sigmoid function of the vertical position between the UAV and the consumer. They then used this model to explain the coverage radius of the UAV as a feature of path loss. They proved how the UAV height could be optimized by maximizing the insurance radius in interference-free surroundings. Moving object recognition and tracking from aerial photos has become one of the most prominent trends in UAV video analysis. Different algorithms have been developed, some of which need GIS data and geo-registration procedures, while others do not. The author used a unipolar restriction and a flow vector bound to identify moving objects. In the research of the study (*Aloqaily et al., 2021*), it is stated that in order to accommodate shifting customer requirements, the article imagines a 5G network scenario in which UAVs powered by blockchain technology provide backup. The technology provides a safe

**Table 1 Existing work and key contributions.**

| Reference | Methodology | Key contribution | Advantage | Disadvantage | Future work |
|---|---|---|---|---|---|
| *Kitchenham (2004)* | Survey and case study | Investigated the potential of unmanned aerial vehicles in smart cities using blockchain. | Improved urban planning and emergency response and readiness. | Limited UAV battery life and coverage range | Develop energy-efficient UAVs |
| *Keele (2007)* | Experimental study | Used unmanned aerial vehicles (UAVs) to monitor traffic using a blockchain-based system. | Data on traffic in real time that is accurate and secure. | High initial setup cost for blockchain | Explore lightweight blockchain protocols |
| *Euchi (2021)* | Simulation and comparative analysis | Blockchain consensus techniques for UAV networks were compared and contrasted. | Greater capacity for business transactions within UAV networks. | Scalability issues with large UAV deployments | Investigate hybrid consensus for UAVs |
| *Lagkas et al. (2018)* | Analytical modeling | Blockchain was used to develop a model that might improve UAV flight paths. | Utilization of available resources in UAV operations in an effective manner. | Limited network bandwidth for UAV communication | Enhance communication protocols for UAVs |
| *Pathak et al. (2020)* | Field experiment and survey | Experiments were carried out in the real-world using UAVs equipped with blockchain technology. | Integrity improvements implemented in the data collected in urban surveys. | Regulatory challenges in drone deployment | Address legal and privacy concerns in UAVs |
| *Sharma et al. (2020)* | System prototype development | Constructed a working model of a system that integrates UAVs and blockchain. | Integrating UAVs smoothly into smart city initiatives. | Integration challenges with existing systems | Develop standardized APIs for UAV integration |
| *Yigitcanlar et al. (2020)* | Comparative study and user feedback | Analyzed the degree to which users were satisfied with blockchain-enabled UAV services. | Satisfaction and participation among the populace increase. | Limited standardization in UAV and blockchain technology | Establish industry standards for UAV services |
| *Gupta et al. (2021)* | Case study and stakeholder interviews | Investigated how stakeholders think regarding unmanned aerial vehicles using blockchain technology. | To learn more about how people feel about UAVs. | Limited public acceptance of UAV technology | Develop public awareness campaigns for UAVs |
| *Mendoza, Rodriguez & Lhuillery (2018)* | Data analysis and machine learning | For the analysis of UAV data, researchers made use of machine learning methods. | Smart city decision-making is enhanced by better data analysis. | Data privacy concerns related to UAV surveillance | Develop privacy-preserving data analytics for UAVs |
| *Stankov et al. (2019)* | Simulation and network optimization | Communication networks for unmanned aerial vehicles that are optimized utilizing blockchain. | Improved communication in terms of both speed and reliability. | Potential security vulnerabilities in blockchain | Implement advanced encryption techniques |
| *Pamnani & Parvathi (2021)* | Field testing and performance evaluation | Researchers conducted tests to see how well UAVs equipped with blockchain technology performed in actual city settings. | Monitoring in real time of metropolitan areas. | Limited scalability due to blockchain complexity | Explore scalable blockchain solutions for UAVs |
| *Alam, Chamoli & Hasan (2022)* | Comparative analysis and cost-benefit study | Using blockchain, researchers analyzed the efficiency of the costs associated with UAVs. | Reduced costs associated with processes while also improving efficiency. | High transaction fees and network congestion | Optimize blockchain parameters for cost efficiency |

and trustworthy means of routing to and from end users and distributing services in a decentralized manner. UAVs use both public and private blockchains to offer access to a broad variety of complicated authorized services and data, with the use of fog and cloud computing devices and data centers. The author in the study (*Aggarwal, Kumar & Tanwar, 2020*) introduce a blockchain-based security solution for UAV communications and 6G network connectivity. The author also provides a brief overview of potential future research avenues regarding the combination of blockchain and 6G technologies for UAV communications. Then, the author provided a case study of how blockchain could be used to secure industry 4.0 applications through UAVs communication over 6G networks. In order to increase the safety of charging transactions, the study of the article (*Qin, Li & Liu, 2021*) investigates the use of blockchain technology to create an auction-based framework for organizing recharging services for UAVs. In this study, the author focusses on improving auction performance by creating a novel auction mechanism based on the upcoming deep reinforcement learning technique, with consideration given to the revenues of carriers and the benefits of UAVs. The experimental outcomes show that the approach can improve the safety of charging transactions and raise the overall efficiency of the system. Table 1 shows the previous work of different authors with methodologies and future work related to the use of UAVs in smart cities.

## METHODOLOGY

### Research questions

The primary objective of this study is to conduct an SLR that identifies, analyzes, and summarizes empirical evidence related to the integration of blockchain with UAVs in smart cities. The review focuses on using blockchain with UAVs in smart cities. It also focuses on the applications, components and types of UAVs. It also focusses on the challenges of UAVs in smart cities. Furthermore, it also focuses on UAVs applications. The article also focuses on the need for the integration of blockchain with UAVs in smart cities make secure transactions. The challenges with the integration of blockchain with UAVs are also discussed and the solutions are also provided in this article. The research questions and the motivation behind each question have been formulated to guide the review process to achieve this goal. Table 2 illustrates the research question, and Fig. 4 shows the proposed methodology.

### Select data sources

Data sources are the libraries or repositories from where the research studies should be retrieved. Four digital libraries have been chosen to extract the primary analyses: IEEE Explore, Science Direct, ACM Digital Library, and Springer Link (*Kitchenham, 2004*). The full text of the documents is searched to identify the prior studies. There are various options available to search each digital library for pertinent information. To find the most relevant literature, the search strategy is modified to satisfy the needs of the respective data source. Selected data sources and the number of studies produced by search queries are illustrated in Table 3 and Fig. 5.

**Table 2 Illustrates the research questions.**

| Research questions | Motivations/Objectives |
|---|---|
| What are the applications and types of UAVs? | To know about the applications and types of UAVs. |
| What are the challenges of UAVs in smart cities? | To know about the different types of challenges of UAV-based systems in smart cities. |
| What are the solutions to key challenges with the of integration blockchain with UAVs? | To know about the solution to these challenges related to blockchain integration with UAVs. |

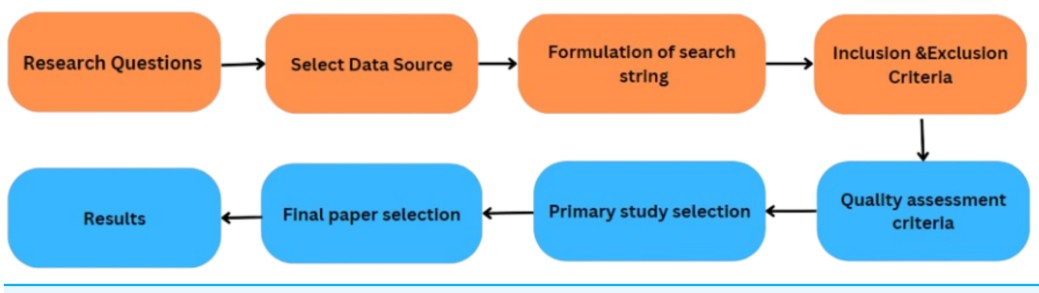

**Figure 4 Proposed SLR.**

**Table 3 Query results from data sources.**

| Library | Initial | Title and keyword | Abstract | Full text |
|---|---|---|---|---|
| ACM | 400 | 250 | 130 | 80 |
| IEEE | 250 | 200 | 90 | 45 |
| Science Direct | 180 | 110 | 75 | 30 |
| Springer | 150 | 120 | 70 | 35 |
| Wiley | 100 | 80 | 45 | 15 |
| Results | 1,080 | 760 | 410 | 205 |

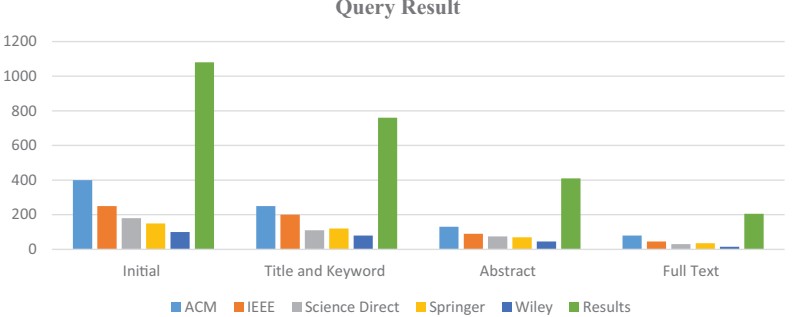

**Figure 5 Search queries representation.**

## Formulate search string

A search string is a carefully crafted combination of keywords and search operators used to identify relevant studies that address the research question or topic of the review. This step focuses on specific keywords and their synonyms chosen from the identified research

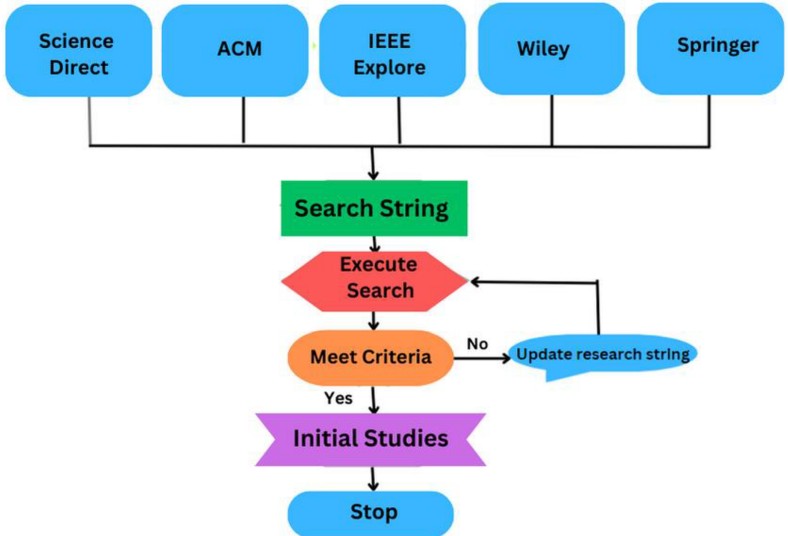

**Figure 6** **The process of formulating the search string.**

**Table 4 Search string formulation.**

| Keyword | Synonym/Alternative word |
|---------|--------------------------|
| Blockchain | ("blockchains" OR "blockchain") |
| UAVs | ("UAVs" *OR* "*Unmanned Aerial Vehicle* Systems") |
| Application | ("uses" OR "types") |
| Methods | ("Techniques" OR "Framework") |
| Integration | ("Combination" OR "merge") |
| Smart cities | ("Big city" OR "Populated area") |

questions, as indicated in Table 2, to create the search string. These keywords are put together using the 'AND' 'OR' conditions in the order listed to complete the following search string: Fig. 6 and Table 4 illustrates the process of formulating a search string.

## Define inclusion and exclusion criteria

- Inclusion criteria in an SLR refer to the predefined rules used to determine which studies will be included in the review. In this review, the following inclusion criteria will be considered:

Studies must have been published in the English language within the timeframe of 2016 to 2023. The subject of the study should be centered on UAVs utilization in the smart city's domain. Selected studies must involve empirical research, conducting practical experiments on specific datasets. The investigations undertaken in the study should pertain to the applications, architecture, and components of UAVs and blockchain. Each chosen study must encompass a comprehensive evaluation of blockchain and

**Table 5 Quality assessment criteria.**

| Sr. No | QA questions |
|---|---|
| C1 | Does the study provide enough information about the applications and types of UAVs? |
| C2 | Does the study provide enough information about challenges of UAVs in smart cities? |
| C3 | Does the study provide enough information about the solutions to Key challenges of integration blockchain with UAVs? |

**Table 6 Final article selection.**

| Year | Final selection |
|---|---|
| 2015 | 1 |
| 2016 | 2 |
| 2017 | 3 |
| 2018 | 12 |
| 2019 | 23 |
| 2020 | 26 |
| 2021 | 21 |
| 2022 | 09 |
| 2023 | 02 |

integration of blockchain with UAVs in smart cities. The scope of selected articles should be confined to publications in reputable journals, conferences, or books.

● Exclusion criteria in an SLR refer to predesigned conditions to determine which studies will be excluded from the review.

The following categories of studies have been designated for exclusion:
Those published before 2016.
Those whose primary focus is not on UAVs and its applications in smart cities.
Studies that lack empirical analysis results.
Studies that fail to evaluate the performance of UAVs and blockchain.

## Define quality assessment criteria

Quality assessment criteria in an SLR refer to the predefined standards or guidelines used to assess the included studies' quality, reliability, and validity. Defining quality assessment criteria ensures that the selected primary studies offer sufficient details to analyze the identified research question effectively. In this step, a standard is defined against each research question. Each quality assessment criterion is denoted by C and its respective number, as shown in Table 5.

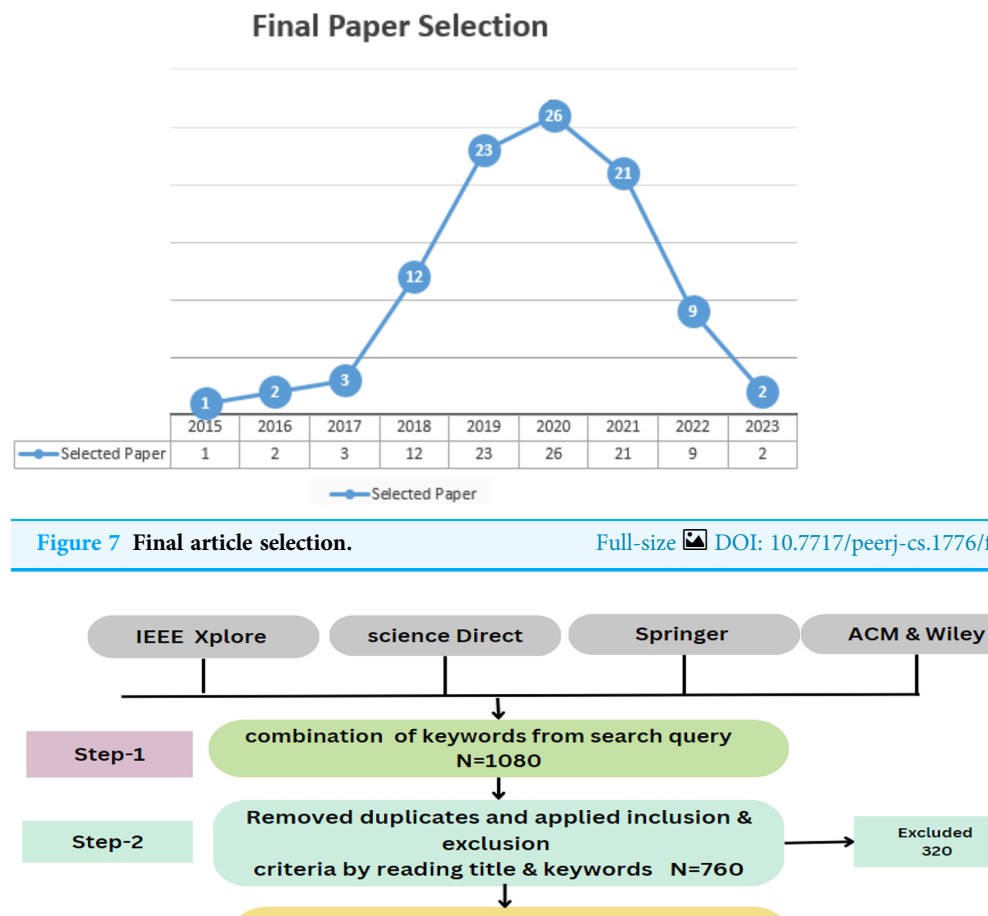

**Figure 7  Final article selection.**

**Figure 8  Prism diagram.**

## Primary study selection

Primary studies refer to the individual articles or book sections that directly address the research questions or topic of the review. This review has selected prior studies using the tollgate approach, a structured methodology of five phases (*Keele, 2007*). This approach was instrumental in carefully curating 49 primary studies, considering the specified quality criteria for prior studies. The primary study selection is illustrated in Table 6 and Fig. 7, and the overall process (prism diagram is shown in Fig. 8).

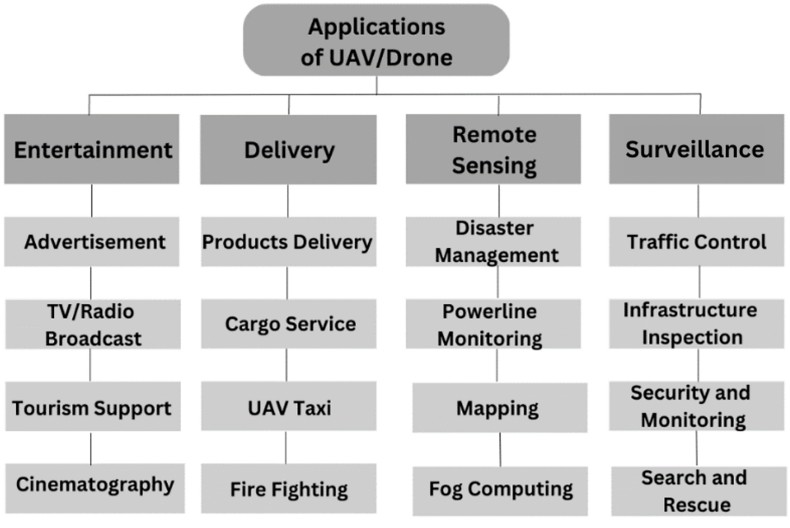

**Figure 9** UAVs applications in smart cities. 

# WHAT ARE THE APPLICATIONS AND TYPES OF UAVS?

## Applications of UAVs in smart cities

Recent advancements in drone technology and related fields have opened up various commercial opportunities. Different drones can be used depending on the task or the issue that must be resolved. Due to emerging scientific technologies, UAVs have become more creative in the public and private sectors (*Euchi, 2021*). UAVs are thought to have a significant advantage in several domains, particularly in public safety, rescue, and product delivery (*Lagkas et al., 2018*). UAVs may potentially be used to bring medical supplies to inaccessible places. UAVs can carry out multiple tasks using sensors, cams, and other payloads (*Pathak et al., 2020*). There are many applications of UAVs in smart cities shown in Fig. 9. UAVs' communication network capacity has been placed to use in various situations. Drones can prove valuable in providing aid in scenarios when other communication channels fail by growing into self-sustaining infrastructure (*Sharma et al., 2020*). This section examines the work done in the deploying of UAV applications in smart cities.

UAV applications significantly deliver various smart city services by optimizing data collection and analysis, increasing infrastructure management, encouraging efficiency, and enhancing emergency service (*Yigitcanlar et al., 2020*).

### Entertainment

The film and entertainment industries have widely used UAVs' high-resolution aerial photos and videography abilities (*Gupta et al., 2021*). The entertainment classification adapts to the average user who enjoys using the latest technological gadgets for taking pictures, aerobatic videos, mapping points of interest, or using a small drone for personal enjoyment. Demand for drone tourism, drone tourists, and the taking of films and images in the air will inevitably increase as the price of drones continues to fall. Smaller drones are employed for agriculture, environmental monitoring, and entertainment (*Mendoza,*

*Rodriguez & Lhuillery, 2018*). Drones have also found a home in entertainment and advertising, allowing them to pull banners and perform light displays. Drone video/photo capturing is more complicated than traditional videography and photography approaches (*Stankov et al., 2019*). UAVs can reach remote and hard-to-access locations. UAVs are used in advertising to capture aerial footage and photos for commercials, product launches, and events. They can also deliver promotional material such as flyers, banners, and promotional products. UAVs can reach places that are difficult to access (*Pamnani & Parvathi, 2021*).

### Delivery

Smart cities' advanced technology and infrastructure are ideal environments for deploying drone delivery systems. In today's modern smart cities, using delivery drones is becoming increasingly common due to their ability to fly above traffic and arrive at their destinations in a fraction of the time (*Alam, Chamoli & Hasan, 2022*). Using drones makes it possible to make supplies faster and more efficient. Drones can access regions hard for ground-based delivery vehicles to reach, such as high-rise buildings and areas restricted to pedestrian traffic only. Because of their superior technology and infrastructure, smart cities are ideally adapted to handle the use of drones, which in turn makes it possible to develop delivery networks that are both effective and efficient (*Watkins et al., 2020*). However, there are also safety and privacy issues surrounding using drones in cities; consequently, appropriate regulations and procedures are required to guarantee that their usage is conducted responsibly and safely. Drones have also become essential in rescue operations, offering a quick and efficient means of accessing disaster-stricken areas and helping those in need (*Novák et al., 2020*).

### Remote sensing

UAVs can deliver products quickly and efficiently, reducing delivery times and improving customer satisfaction (*Kim, Moon & Jung, 2020*). UAVs uniquely provide high-quality data from remote sensing at this scale and timeframe (*Maes & Steppe, 2019*). The footage captured by UAVs enables hazards and features to be observed from various angles in virtually any landscape, which is impossible with any other single technology (*Nikhil et al., 2020*). UAV taxi services are being developed and tested by several companies worldwide. Vole Copter has developed a two-seat electric air taxi with a range of 60 km, which can be charged and provide faster transportation, reducing travel times and improving accessibility (*Marzouk, 2022*). During the tests performed in 2017, a Chinese drone taxi called Ehang-184 flew successfully in Dubai City (*Alnuaimi, 2021*).

UAVs play a vital role in surveying the affected area and assessing the damage caused by a disaster. UAVs can deliver emergency supplies such as food, medicine, and medical equipment to disaster-stricken regions (*Mohsan et al., 2022b*). UAVs play an essential role in power line monitoring in smart cities. UAVs can detect cracks, corrosion, or loose connections before they lead to power outages or other problems. Inspections can be performed without putting human inspectors at risk, especially in hazardous environments (*Ciampa, Vito & Pecce, 2019*).

**Table 7  An overview and summary of UAV applications for smart cities and their possible benefits.**

| UAV application | UAV function | Possible benefits |
|---|---|---|
| Advertisement | Used in advertising to capture aerial footage and photos for commercials product. | UAVs can reach places that are difficult to access. |
| UAV-based tourism | Involve more people in digital tourism. | UAVs can reach remote and hard-to-access locations. |
| Security and monitoring | Assure public safety and security. | UAVs can monitor large areas, such as borders, coastal regions, and industrial sites. |
| UAV-based cinematography | UAVs can fly in tight spaces. | Enhance the visual impact of a film. |
| UAV-based product delivery | UAVs can deliver products quickly. | Reduce delivery times and improve customer satisfaction. |
| Traffic control | Provide a unique viewpoint on traffic monitoring. | Assist in the optimization of road traffic systems. |
| Cargo services | Deliver a wide range of cargo. | It can help to improve supply chain operations by providing real-time tracking. |
| Fire fighting | Improve firefighting response and conduct rescue operations. | Drones can visit areas that are tough to get to and carry out data collection tasks that are hard for people to do. |
| UAV-based taxi | Provide faster transportation. | Improving accessibility. |
| Fog computing | Optimize energy usage in smart cities. | Provides real-time monitoring of energy consumption. |
| Disaster management | Deliver emergency supplies. | Play a vital role in surveying the affected areas and assessing the damage caused by a disaster. |
| Infrastructure inspection | Inspect and assess the condition of infrastructure. | Equipped with high-resolution cameras and sensors can access hard-to-reach areas and provide real-time data and images. |
| Search and rescue | Conducting rescue operations. | More efficient than traditional manned-held operations. |
| Power line monitoring | Play an essential role in power line monitoring in smart cities. | Inspections can be performed without putting human inspectors at risk, especially in hazardous environments. |

## Surveillance

With recent advancements, UAVs are seen as a possible alternative to ground robots for surveillance in various applications. UAV surveillance can move faster than ground vehicles or robotics, resulting in quicker turnaround times between the desired locations (*Gu et al., 2018*). The traffic monitoring system is one of the essential applications for UAVs. Due to their mobility and ability to cover a wide variety of areas, drones provide a unique viewpoint on traffic monitoring that has the potential to assist in the optimization of road traffic systems. This could be achieved by exceeding conventional surveillance methods (*Khan et al., 2020b*). These drones also check industrial sites and power transmission infrastructure (*Ramon-Soria et al., 2019*). *Yu et al. (2019)* Suggested solutions for localizing UAVs while inspecting pipelines. UAVs have been used to examine power lines using similar concepts (*Ramon-Soria et al., 2019*). UAV-based crowd-monitoring systems are feasible and cost-effective solutions, making them a vital component of technology for crowd-monitoring (*Xiao et al., 2021*). UAVs can monitor large areas, such as borders, coastal regions, and industrial sites, for security purposes. They can quickly reach an incident scene to gather information and assess the situation, allowing faster response time (*Alsamhi et al., 2021*). UAV-based infrastructure inspection is a cost-

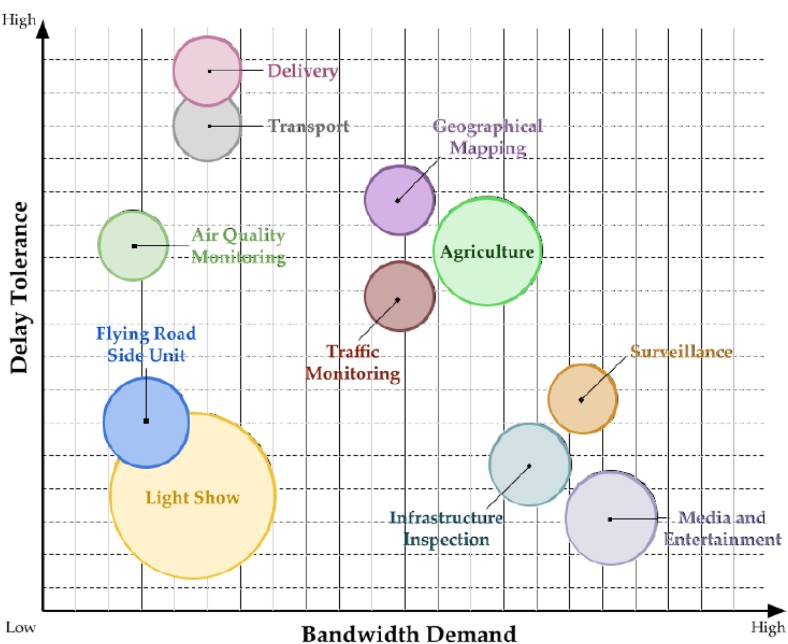

**Figure 10 Highlighting diverse UAV applications for smart cities.**

effective and efficient method to inspect and evaluate the condition of infrastructure such as buildings, bridges, pipelines, and power lines. UAVs with high-resolution cameras and sensors can access hard-to-reach areas and provide real-time data and images (*Macaulay & Shafiee, 2022*). Table 7 shows an overview and summary of UAV applications for smart cities and their possible benefits.

Ensuring the availability of fundamental life-saving and emergency healthcare services is crucial for every city. Cities must provide these services to their residents regardless of their location or the time of need. UAVs present a viable solution for delivering or assisting health emergency services. UAVs can swiftly transport essential medical supplies and services to patients, enabling the rapid availability of basic life support systems and healthcare provisions. For instance, a UAV can efficiently transport a defibrillator and other necessary medical equipment directly to a cardiovascular patient, facilitating immediate cardiopulmonary resuscitation.

### Traffic management and monitoring

Traffic congestion is a significant issue that confronts numerous major cities. It arises from various factors, such as peak hours, major events, construction activities, or accidents, leading to sudden or periodic increases in the number of vehicles. These issues can arise at any time and in any city location. Consequently, transportation administrators must seek more efficient and effective approaches to alleviate traffic congestion. To identify an appropriate solution, it is crucial to know congestion-prone areas, their causes, the volume of vehicles, and the condition of the roads in those areas. Although static cameras installed on city streets can offer some insights into these factors, they often fail to provide

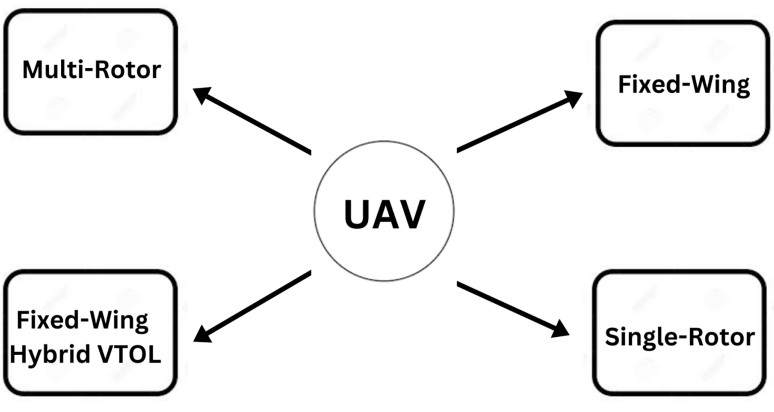

**Figure 11 Types of UAV.**

comprehensive real-time information necessary for managing the problem (*Chen et al., 2016*). Figure 10 highlights diverse UAV applications for smart cities (*Lucic et al., 2023*).

*Agriculture management and environmental monitoring*
UAVs can support various agricultural processes, including distributing fertilizers, pesticides, seeds, and water. They also serve the purpose of conducting regular inspections of crops and assessing their overall condition. Additionally, UAVs can assist in monitoring the growth of crops to determine optimal harvesting times or the need for protective measures. For instance, certain crops are susceptible to significant damage due to sudden weather changes. UAVs are instrumental in monitoring and measuring environmental conditions, such as identifying high levels of harmful substances like $CO_2$ and air pollutants. This early detection capability enables timely action and accurately assesses the associated risks (*Mohamed et al., 2020*).

## Types of UAVs

UAVs drone, also known as a drone, is operated without a human pilot (*Chaurasia & Mohindru, 2021*). They use autonomous or remote-control systems to fly, making them suitable for a wide range of applications, including aerial photography, delivery of goods, an inspection of infrastructure, and military operations. A UAV drone typically consists of a flight control system, a propulsion system, navigation sensors (such as GPS), communication systems, and payload sensors or cameras (*Alghamdi, Munir & La, 2021*). The flight control system receives input from the remote operator or autonomous onboard systems and then controls the drone's movement through its propulsion system. Navigation sensors help the drone determine its position and orientation, while communication systems allow it to receive commands and transmit data (*Sandino et al., 2020*). The payload sensors or cameras collect data or capture images and video. UAVs use various technologies to operate effectively and efficiently (*Albeaino, Gheisari & Franz, 2019*).

The flight mission of a UAV is typically predefined, but a pilot can manage its velocity and direction *via* remote robotic system commands from a base station (*Radoglou-Grammatikis et al., 2020*). UAVs are available in a variety of sizes, features, and models.

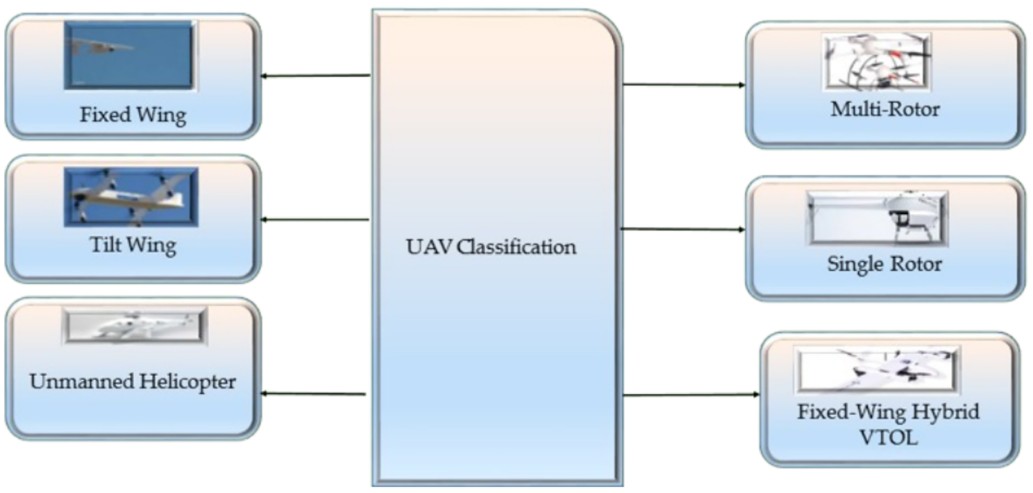

**Figure 12** Shows different classifications of UAVS.

UAVs of various combinations are employed for a variety of reasons. We distinguish and analyze multiple types of UAVs in this area according to their technical attributes and payload. Figure 11 shows the four major types of UAVs. Figure 12 (*Partheepan, Sanati & Hassan, 2023*) shows different classifications of UAVs.

### Multi-rotor

Multi-rotor UAVs are predicted to become increasingly intelligent in the future. Multi-rotor drones, often rotary-wing drones, are the most common leisure and professional drones (*Green & Gómez, 2020*). Multi-rotor drones, also known as quadcopters, are a type of UAV popular in smart city applications due to their agility, stability, and versatility. They provide significant adaptability by enabling the installation of several kinds of cameras to execute various tasks. Because of their outstanding mobility, strength, and hover performance, multi-rotor UAVs have seen widespread use in public and military marketplaces in recent years (*Quan, Dai & Wang, 2020*). Multi-rotor drones equipped with cameras and sensors can be used for surveillance, helping to increase public safety and reduce crime (*Abichandani, Levin & Bucci, 2019*). UAVs can capture stunning aerial footage, providing a unique perspective and enhancing the visual impact of a film (*Patel, 2019*). Increasing the number of rotors makes the drone more challenging to control. The majority of multi-rotor drones have flying times of less than one hour. The great sensitivity of multi-rotor UAVs to wind disruptions is a significant challenge in their use.

### Fixed-wing

Fixed-wing drones have a rigid structure, similar to an aero plan, with lift wings, enabling the drone to fly in the air. Fixed-wing drones are typically used for various applications, including aerial photography, surveying and mapping, inspecting infrastructure and assets, delivering goods, and scientific research (*Lee, Kim & Chu, 2021*). They can fly for extended periods and cover larger areas than rotary-wing drones, making them well-suited for tasks requiring high stability and efficiency. Regarding capacity, fixed-wing drones can carry a

variety of payloads, including cameras, sensors, and other equipment, depending on their design and specifications (*Green et al., 2019*). They can also be designed to fly at different altitudes and speeds, allowing them to capture high-resolution images and data in various conditions (*Ziliani et al., 2018*).

### Single-rotor

Drones with a single rotor are driven by a single main rotor, responsible for providing lift, and a tail rotor, responsible for providing directional control. The drone can get off the ground and stay in the air (*Zhang et al., 2018*). At the same time, the tail rotor is responsible for providing stability and directionality by counteracting the torque generated by the main rotor. Drones with a single rotor rely on the interaction between their primary and tail rotors to develop lift, keep their stability, and adjust the directionality of their flying (*Khan et al., 2021b*). The flight controller is the "brain" of the drone. It is responsible for controlling the movement of the rotors and maintaining the drone's stability while in the air by using information from the operator or the GPS (*Mogili & Deepak, 2018*).

### Fixed-wing hybrid VTOL drones

Fixed-wing hybrid VTOL drones can take off, land vertically like a helicopter, and fly horizontally like fixed-wing Aerial Vehicle. These drones typically have fixed wings and rotors to transition between vertical and horizontal flight (*Zhou et al., 2020*). In vertical takeoff mode, the rotors lift the drone, allowing it to take off and hover. Once the drone has reached a certain altitude, it can transition to horizontal flight mode, in which the fixed wings provide lift and the rotors are used for propulsion (*Li et al., 2020*). During horizontal flight, the drone is much more efficient and can cover greater distances than it could with rotors. Their efficiency during horizontal flight allows them to cover greater distances and fly longer than traditional rotary-wing drones (*Tkáč & Mésároš, 2019*).

## WHAT ARE THE CHALLENGES IN UAVS IN SMART CITIES?

While UAVs hold the potential to provide numerous advantages for smart city services, their effective utilization entails addressing several challenges. These challenges can be broadly classified into two categories: technical issues, such as safety, security, reliability, and communication, and non-technical issues, such as licensing and legislation, cost, ethics, and privacy (*Mohamed et al., 2020*).

Several benefits are associated with UAVs, many of which have increased significantly as technology advances. However, some have limited resources for operation, while others present various security challenges. A lack of standardization and regulation of UAVs leads to security and privacy concerns. Due to design constraints and a hostile operational area, security specialists have problems constructing a complete UAV security module (*Mugheri, Siddiqui & Khoso, 2018*). Although UAVs have the potential to provide numerous benefits for the services offered in smart cities effectively, however, using UAVs also presents various challenges that need to be overcome. Herein the list of challenges in UAVs in smart cities is discussed.

## Speed and security

UAV drones have the potential to revolutionize delivery services by offering faster and more efficient delivery times. However, drones' speed can also threaten public safety if operated at high speeds in densely populated areas (*Cherif et al., 2021*). UAV drones also raise security concerns, particularly in surveillance and privacy. Drones equipped with cameras and other sensors can be used to collect sensitive data, and there is a risk of drones being hacked and used for malicious purposes (*Nassi et al., 2019*). It is essential to address speed and security concerns through effective regulations and best practices to ensure their safe and efficient integration into urban environments.

## Secure communication channel

There is a need for a secure communication channel for UAVs in smart cities to protect against various security threats and ensure drones' safe and efficient operation. A secure communication channel can help protect this data from unauthorized access and ensure it is transmitted and stored securely (*Nikooghadam et al., 2021*). UAVs must be operated safely to avoid accidents and other incidents that could risk public safety. Secure communication can help ensure that UAVs are managed, controlled, and safe by providing real-time communication and control between the drone and its operator (*Khan et al., 2020a*).

## Inventory management system

Strict regulations surround the use of drones, especially regarding their usage in commercial operations. Most drones have limited flight time, which can challenge extended inventory management tasks (*Reddy Maddikunta et al., 2021*). Drones have limited storage capacity, making transporting and storing large items challenging for inventory management purposes. Operating and maintaining drones for inventory management requires technical expertise and training, which can be challenging for some organizations (*Wawrla, Maghazei & Netland, 2019*).

## Global channel for connectivity and communication

Global communication channels are necessary for UAVs in smart cities to ensure coordination, safety, and efficiency in air traffic management. With the increasing use of drones for various purposes, it is essential to have a unified communication system that allows for real-time communication and exchange of information between different drone systems, ground control stations, and air traffic control (*Vinogradov et al., 2019*). This helps prevent potential collisions, ensure regulations comply, and allow for optimized flight paths. Global communication channels also facilitate communication with other air and ground-based vehicles and can provide critical information such as weather conditions, air traffic updates, and emergency alerts (*Al-Mousa et al., 2019*).

## Inter-service operation capabilities

Inter-service operation capabilities in drones refer to the ability of drones to interact and coordinate with other technologies, systems, and services in smart cities. Different services

may use different communication systems, leading to poor interoperability (*Aloqaily et al., 2021*). UAVs collect vast amounts of data, but sharing that data between various services can be challenging. Different services may have other operating procedures, leading to confusion and misunderstandings during UAV operations (*Zhai et al., 2020*). These challenges can limit the effectiveness and efficiency of UAV operations, making it essential for services to work together to overcome them.

## Limited storage and processing power

The amount of storage and onboard processing power limits UAVs. These limitations can affect the performance and capabilities of the drone. Drones also have limited processing power, impacting their ability to process data in real time, especially for complex tasks such as object recognition, obstacle avoidance, and real-time decision-making (*Fraga-Lamas et al., 2019*). Drones' limited storage and processing power can also impact their flight time, as these systems consume battery power, reducing the time the drone can remain in the air (*Boukoberine, Zhou & Benbouzid, 2019*).

## Autonomous network security system

There are several challenges in implementing an autonomous network security system for UAVs in smart cities. UAVs operate in the same frequency bands as other communication systems, which can lead to interference and disruption of normal operations (*Li et al., 2021*). As UAV usage grows, the communication system must be able to handle increasing numbers of devices and provide adequate coverage to accommodate changing traffic patterns. UAVs require significant bandwidth, which can be challenging to allocate and manage efficiently (*Khan et al., 2021a*).

## Securely sending data to end users

Data transmitted between UAVs and end users can be intercepted by unauthorized parties, compromising security. UAVs generate large amounts of data, leading to network congestion, particularly in densely populated urban environments (*Alam et al., 2021*). Other wireless communication systems in the urban environment can interfere with UAV communication, leading to errors and data loss. The privacy of end-user data must be protected, and UAV communication systems must ensure that data is transmitted securely and encrypted to prevent unauthorized access (*Atoev et al., 2019*).

## Blocking of the line of sight

Blocking the line of sight is a common challenge in UAV communication in smart cities due to tall buildings, bridges, and other structures that can obstruct the communication between the UAV and the ground station or end-user (*Yu et al., 2022*). The UAV may lose contact with the ground station or end user, impacting real-time data transmission (*Mohsan et al., 2022a*). Obstructed line of sight (LoS) can increase latency and reduce the reliability of the UAV communication system. UAV communication systems may not provide adequate coverage in areas with obstructed LoS, leading to dead communication zones (*Albanese, Sciancalepore & Costa-Pérez, 2021*).

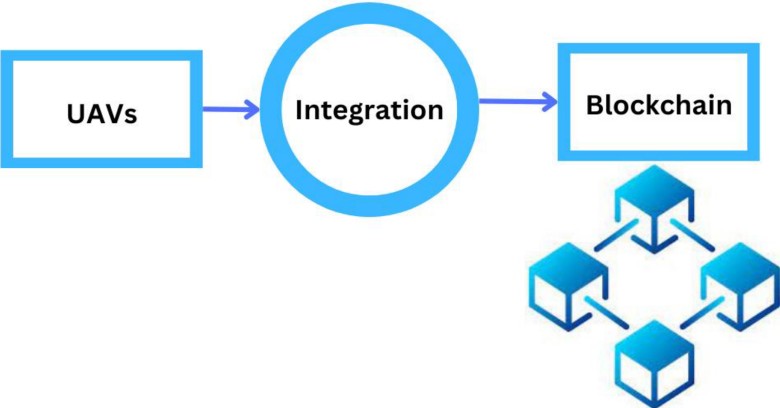

**Figure 13** **UAVs integration with blockchain.**

## UAV surveillance network

UAVs must be able to cover a large area in real time, which requires a robust and scalable communication network. UAVs have limited energy resources, and the communication and data processing systems must be designed to minimize power consumption (*Zhang et al., 2020*). UAV surveillance systems must protect individuals' privacy and comply with relevant privacy regulations. This surveillance system must be integrated with existing surveillance and security systems to ensure efficient and effective operations (*Lykou, Moustakas & Gritzalis, 2020*).

## UAV networks for edge computing

UAV networks for edge computing can be used in various applications, including surveillance, environmental monitoring, agriculture, and emergency response (*Xu, Ota & Dong, 2020*). UAV networks for edge computing involve using a network of UAVs to collect and process data locally rather than relying on a centralized computing system (*Liu et al., 2022*). This enables real-time decision-making and reduces the amount of data that must be transmitted over long distances, thus reducing latency and increasing data processing speed. As the number of UAVs in a network grows, it becomes more challenging to manage and coordinate communication and data processing among the devices (*Alzahrani et al., 2020*).

## Limited battery issue

Depending on factors such as its size, cargo, and battery life, a UAV's flying time may be as little as 20 min or as long as several hours. A substantial quantity of energy is required to complete these duties. Additionally, not all drones have sufficient battery power to provide services for a lengthy duration of time (*Islam et al., 2021*). UAVs have a significant difficulty in the context of smart cities due to their limited battery life (*Dai et al., 2021*). As UAVs grow more commonplace in urban settings for purposes like surveillance, traffic monitoring, environmental evaluations, and emergency response, the problem of limited battery capacity becomes all the more pressing.

# WHAT ARE THE SOLUTIONS TO KEY CHALLENGES WITH THE INTEGRATION OF BLOCKCHAIN AND UAVS?

Blockchain is a decentralized digital ledger that uses cryptography to secure and record transactions across a network of computers. Blockchain technology can offer new solutions in various aspects when combined with drones, as described in Fig. 13.

In smart cities, blockchain technology can solve several issues related to UAVs. Blockchain can create a decentralized traffic management system for UAVs, allowing for real-time drone operation coordination and reducing collision risk (*Albanese, Sciancalepore & Costa-Pérez, 2021*). Blockchain can facilitate rapid and secure communication between UAVs and emergency responders, allowing faster and more efficient response times in case of accidents or disasters. It can also create a safe and tamper-proof record of UAV sensor data, allowing for more effective monitoring of traffic, pollution, and other environmental factors in smart cities (*Zhang et al., 2020*). Blockchain ensures the privacy and security of data collected by UAVs, enabling the secure sharing of data among authorized stakeholders. Blockchain-based smart contracts can allow autonomous and safe package deliveries without human intervention and provide transparency on the delivery status and location (*Lykou, Moustakas & Gritzalis, 2020*). This technology can support the development of new use cases for UAVs across various industries, such as logistics, transportation, and public safety. Table 8 presents the different proposed solutions, which are Blockchain-based solutions, to solve the problems in UAVs in smart cities.

## Blockchain integration with UAVs for better speed and security

The combination of UAVs and blockchain technology has the potential to improve efficiency and safety in many situations. Blockchain technology allows for decentralized and immutable data storage. Images, sensor readings, and flight logs from UAVs can all be recorded in a distributed ledger. The complete supply chain of UAV components could be tracked with blockchain technology. By eliminating the possibility of using fake or substandard components, this improves the UAVs' dependability and security. In the study of *Ch et al. (2020)* the research introduces a blockchain technology (BCT)-based approach to protecting the confidentiality of information collected by devices. To test the effectiveness of the proposed design, an IoT based application has been integrated into a simulated vehicle monitoring system. Penta tope-based elliptic curve encryption and Secure Hash Algorithm (SHA) are used to secure privacy in data storage, and this includes the technical details of instructions to the vehicle (devices), authentication, integrity, and vehicle reactions. In order to facilitate BCT transactions, the information is later recorded in a public blockchain based on Ethereum. The system relies on the secure and private Ganache platform for BCT to store and process user information.

## Blockchain integration with UAVs for secure communication channel

Blockchain technology can be utilized for the preservation of encryption keys. The blockchain can be used to encrypt messages transferred between UAVs, making it so that only approved UAVs can read them. With the use of blockchain technology, UAVs could

**Table 8 Blockchain-based solutions to challenges of UAVs in smart cities.**

| Sr. # | Application domain | Problem addressed | Proposed blockchain solution | Reference |
|---|---|---|---|---|
| 1 | Speed and security. | The speed of drones can also pose a risk to public safety if they are operated at high speeds in densely populated areas | a Blockchain Technology (BCT)-based approach to protecting the confidentiality of information collected by devices. To test the effectiveness of the proposed design, an IoT-based application has been integrated into a simulated vehicle monitoring system. | Zekiye & Özkasap (2023) |
| 2 | Secure communication channel. | There is a need for a secure communication channel for UAVs in smart cities to protect against various security threats and ensure drones' safe and efficient operation. | With its public and private key processes, Blockchain can serve as a secure communication channel among UAVs. | Ghribi et al. (2020) |
| 3 | Inventory management system. | Drones have limited storage capacity, made transporting and stored large items challenging for inventory management purposes. | A Blockchain-based inventory management system that uses UAVs to scan items equipped with RFID tags and then uploads that data to a Blockchain, where it is validated, and transparency is maintained. Smart contracts are utilized when doing business with third parties to facilitate the transaction. | Cristiani et al. (2020) |
| 4 | Global channel for communication. | Global communication channels are necessary for UAVs in smart cities to ensure coordination, safety, and efficiency in air traffic management. | Employing Blockchain as a worldwide communication platform, UAVs are equipped with the capacity to sign and send encoded data securely. It enables UAVs in the network to decide transparently by allowing them to consider the viewpoints of other UAV users. | Kumar et al. (2021) |
| 5 | Inter-service operation capabilities. | UAVs collect vast amounts of data, but sharing that data between different services can be challenging. | A trust agreement between vendors is built on Blockchain technology to facilitate inter-service activities, with each UAV operating as a node in the Blockchain. Because the record of services given by each vendor is stored on the Blockchain in an accessible and open way, this facilitates an environment of trust and confidence among all parties involved. | Sharma, You & Kul (2017) |
| 6 | Limited processing power and storage. | The amount of storage and onboard processing power limits UAVs. These limitations can affect the performance and capabilities of the drone. | A Blockchain-powered decentralized storage system in which UAVs that serve as air sensors pass their data to ground sensors and, in exchange, compensate the ground sensor with incentives dispersed via Blockchain for their storing and processing abilities. | Zhu, Zheng & Wong (2019) |
| 7 | Autonomous network security system. | UAVs operate in the same frequency bands as other communication systems, which can lead to interference and disruption of normal operations. | A Blockchain-based network for UAVs in which each UAV carries a copy of the onboard Blockchain. If command signals from the control center or other network UAVs are disrupted, UAVs can consult the data stored on Blockchains to determine their flight paths and action plans. | Alkadi et al. (2022) |
| 8 | Securely sending data to end users. | Data transmitted between UAVs and end users can be intercepted by unauthorized parties, compromising its Security. | Protecting the data gathering and transmission processes in an Internet of drone environment, the author used a public Blockchain based on Ethereum. It guarantees the data's honesty, accountability, authorization, and privacy. | Aggarwal et al. (2019) |
| 9 | Blocking of the LoS. | Blocking LoS is a common challenge in UAV communication in smart cities due to tall buildings, bridges, and other structures that can obstruct communication between the UAV and the ground station or end user. | A Blockchain-based UAV traffic data exchange network is currently in development to facilitate the safe transfer of traffic data. Using Blockchain, it is possible to develop a double-block verification mechanism, rendering the system resistant to cyber-attacks and blockages in the Los. | Chao et al. (2018) |
| 10 | UAV surveillance network. | UAV surveillance systems must protect individuals' privacy and comply with relevant privacy regulations. | Block-chain-based security strategies can identify suspicious occurrences in the surveillance data and fraudulent UAVs through a distributed trust management strategy. | García-Magariño et al. (2019) |

(Continued)

| Table 8 (continued) | | | | |
|---|---|---|---|---|
| Sr. # | Application domain | Problem addressed | Proposed blockchain solution | Reference |
| 11 | UAV networks for edge computing. | As the number of UAVs in a network increases, managing and coordinating communication and data processing among the devices becomes more challenging. | An approach to providing ultra-reliability based on the Blockchain and neural networks and uses UAVs as on-demand nodes for caching objectives. | *Sharma et al. (2019)* |
| 12. | Limited battery issue. | UAVs have a significant difficulty in the context of smart cities due to their limited battery life. | A blockchain-based energy grid can store surplus energy generated from renewable sources, allowing for a decentralized energy management system | *Zekiye & Özkasap (2023)* |

safely authenticate each other's identities before opening up communication lines. UAVs used for delivery services can benefit from blockchain's ability to track the entire supply chain, which in turn assists in preventing counterfeiting. As the network grows larger, the use of blockchain technology could assist in guaranteeing its security and transparency. In order to ensure the safety of communications in a UAV network, the authors of the study (*Ghribi et al., 2020*) suggest a new consensus-building methodology that combines blockchain technology, the Elliptic Curve Diffie-Hellman public key cryptography technique, and the one-time pad encryption technique.

## Blockchain integration with UAVs for inventory management system

UAVs equipped with sensors and GPS can monitor shipments in real time. With the use of the blockchain, all parties involved may track the whereabouts and status of stock in real time. Using low-cost mini-drones and single-board computers, The research of *Cristiani et al. (2020)* tested a prototype implementation of the data acquisition and management framework. Our goal is to propose a generic architecture for UAV-based inventory management inside large-scale warehouses, including the elements of UAV path planning, package recognition (*via* QR codes), data validation (*via* the blockchain), and wireless charging. Second, authors examine the system's effectiveness, focusing on the compromise between inventory accuracy (the proportion of correctly identified packages) and total inventory processing time. Finally, using a small-scale testbed, the author validates the system's functionality and configuration parameters by determining the ideal UAV mobility characteristics with regard to of speed and number of visits for each shelf unit.

## Blockchain integration with UAVs for global channel for connectivity and communication

Using cryptographic techniques like public-key encryption and hash functions, blockchain ensures the integrity of the distributed ledger. It can be used to improve the accessibility and safety of aerial communication networks while also protecting the integrity of stored data. The research of *Kumar et al. (2021)* gives the results of an online survey on the topic of using blockchain technology with BAC. They begin by investigating the existing state of security in aerial communication networks, the benefits of blockchain technology, and the

potential for its use in a solution. The next section provides an in-depth discussion of the author's suggested options for implementing the blockchain in order to address the existing security vulnerability in aerial communication networks. Finally, the authors sort the solutions into categories and assess and contrast the benefits and drawbacks of each.

## Blockchain integration with UAVs for inter-service operation capabilities

End-to-end encryption of UAV-based service-to-service communication channels is made possible by blockchain technology. The public and private keys for each service can be maintained in the distributed ledger to protect user data and maintain confidentiality. Cooperation between government bodies, technologists, and legal experts is necessary to integrate blockchain technology with UAVs to enable inter-service operating capabilities. When demand is too high for a single service provider to handle, blockchain can be used to coordinate the actions of multiple providers' UAVs to meet that demand. UAVs' operability in such a context is examined. In the study of *Sharma, You & Kul (2017)* a trust agreement between vendors is built on blockchain technology to facilitate inter-service activities, with each UAV operating as a node in the blockchain. Because the record of services given by each vendor is stored on the blockchain in an accessible and open way, this facilitates an environment of trust and confidence among all parties involved.

## Blockchain integration with UAVs for limited storage and processing power

Blockchain technology can be successfully implemented in environments with limited resources such as storage and computing power. The architecture developed by *Liang et al. (2017)* ensures the authenticity of drone data by combining public blockchain technology with cloud storage. Performance evaluation results are acceptable after implementing a prototype of a drone system. The study of the author *Zhu, Zheng & Wong (2019)* proposed a blockchain-powered decentralized storage system in which UAVs acting as air sensors transmit data to ground sensors, which are then rewarded with incentives distributed *via* Blockchain for their capacity to store and interpret the data.

## Blockchain integration with UAVs for autonomous network security system

The safety and efficacy of autonomous network systems can be greatly improved by incorporating blockchain technology with UAVs. In applications where UAVs operate independently and require secure communication with a centralized network, such as disaster response or surveillance, this integration can prove invaluable. A blockchain-based network in which each UAV has its own copy of the blockchain is proposed as a solution in the research (*Alkadi et al., 2022*). Data recorded on blockchains can be used by UAVs to figure out their flight paths and actions in the event that they lose contact with the control center or other network UAVs.

### Blockchain integration with UAVs for securely sending data to end users

Data sent from UAVs to their ultimate recipients can be encrypted using blockchain technology. In order to protect the privacy and security of the information recorded on the blockchain, it might be encrypted before it is added. Micropayments for the data collected by UAVs are possible with blockchain-based solutions. An incentive for UAV operators to collect and communicate valuable data to end users in a secure manner is provided by smart contracts, which can automatically execute payments when data is delivered and validated. The author *Aggarwal et al. (2019)* used a public Ethereum-based blockchain to examine how to secure data collection and transmission in an Internet of Drones setting. It ensures the truthfulness, traceability, authorization, and confidentiality of the data.

### Blockchain integration with UAVs for blocking of the line of sight

Blockchain technology was developed largely for use in managing and transacting data in a way that is both safe and transparent. It does not interfere with physical processes, such as obstructing the view of UAVs. However, when used in tandem with other technologies and protocols, blockchain can improve the safety and management of UAVs flying under limited or obstructed Line of Sight conditions. In the study of *Chao et al. (2018)* a safe and reliable data sharing amongst UAVs is being developed through the use of a Blockchain-based network. A double-block verification process can be created using Blockchain, making the system immune to cyberattacks and bottlenecks in the Los.

### Blockchain integration with UAVs for UAV surveillance network

Integrating blockchain technology with UAVs for a surveillance network can boost the security, transparency, and efficiency of data handling. The security and trustworthiness of surveillance data can be guaranteed by using blockchain, which has various benefits in this context. Access control in a surveillance network can be managed with the help of smart contracts. Based on the conditions recorded in smart contracts, authorized users, such as law enforcement agencies or security staff, can gain access to specific surveillance data. A technique is represented in the research of *García-Magariño et al. (2019)*. Security and personal privacy are both improved by this system of automatic access management. Through a distributed trust management technique, block-chain based security solutions may detect anomalies in surveillance data as well as fraudulent UAVs.

### Blockchain integration with UAVs for UAV networks for edge computing

In the context of UAV networks for edge computing, incorporating blockchain technology with UAVs can produce novel and secure applications. Instead of depending only on remote cloud servers, edge computing moves data processing closer to its point of origin. In this way, data can be processed and stored independently using blockchain technology. In the research of *Sharma et al. (2019)* it is shown that UAVs can do data processing locally, allowing them to make decisions in real time at the network's edge. UAVs are used

as on-demand nodes in a distributed blockchain network powered by neural networks to achieve caching goals.

### Blockchain integration with UAVs for limited battery issue

The low battery life of UAVs is not something that can be fixed by blockchain technology on its own. However, the limitations of battery life can be indirectly addressed by integrating blockchain into UAV systems in smart cities to optimize their operations. Blockchain-based smart contracts can run algorithms that optimize UAV routing, ensuring that drones travel the most efficient routes possible, and hence extending their runtime. A blockchain-based energy grid can store surplus energy generated from renewable sources, allowing for a decentralized energy management system (*Zekiye & Özkasap, 2023*).

## CONCLUSION AND FUTURE DIRECTION

UAVs in smart cities open new possibilities and bring cost-effective intelligent solutions to various challenges. Several of these possibilities will result in incredibly useful applications such as intelligent traffic management, smart safety and security, smart entertainment, smart health care, and agriculture management. Most existing and future applications can considerably benefit from incorporating UAVs into smart cities. Smart city researchers are looking at blockchain technologies for managing, analyzing, and addressing the security and other challenges associated with UAV implementation. The application and types of UAVs in smart cities are summarized in this article, focusing on the problems of UAVs in smart cities and why blockchain approaches are vital for addressing these challenges. Using UAVs in smart cities offers various advantages but has serious limitations. We demonstrated the use of UAVs in smart cities and discussed the challenges associated with UAVs. Blockchain can address these challenges in smart cities. This article has examined the most popular blockchain technologies used in smart cities to handle various security issues. These blockchain approaches can be extremely useful with UAVs in a smart city. There are still several challenges in the use of UAVs in smart cities. However, recent technical advancements in blockchain technology have enabled the concept of UAVs to be used in smart cities. Suppose we can find more solutions to the many challenges on time. In that case, it will greatly motivate advancements in smart city developments' academic and industry realms.

### Funding
The authors received no funding for this work.

### Competing Interests
Habib Hamam is the director of Spectrum of Knowledge Production and Skills Development, Sfax, Tunisia.

## Author Contributions

- Syed Faisal Abbas Shah conceived and designed the experiments, performed the experiments, analyzed the data, performed the computation work, prepared figures and/ or tables, authored or reviewed drafts of the article, and approved the final draft.
- Tehseen Mazhar conceived and designed the experiments, performed the experiments, analyzed the data, performed the computation work, prepared figures and/or tables, authored or reviewed drafts of the article, methodologist, and approved the final draft.
- Tamara Al Shloul conceived and designed the experiments, performed the experiments, analyzed the data, performed the computation work, prepared figures and/or tables, authored or reviewed drafts of the article, software, and approved the final draft.
- Tariq Shahzad conceived and designed the experiments, performed the experiments, analyzed the data, performed the computation work, prepared figures and/or tables, authored or reviewed drafts of the article, resources, and approved the final draft.
- Yu-Chen Hu conceived and designed the experiments, performed the experiments, analyzed the data, performed the computation work, prepared figures and/or tables, authored or reviewed drafts of the article, and approved the final draft.
- Fatma Mallek conceived and designed the experiments, performed the experiments, analyzed the data, performed the computation work, prepared figures and/or tables, authored or reviewed drafts of the article, visualization, Methodologist, and approved the final draft.
- Habib Hamam conceived and designed the experiments, performed the experiments, analyzed the data, performed the computation work, prepared figures and/or tables, authored or reviewed drafts of the article, project administration, funding acquisition, and approved the final draft.

## Data Availability

This is a literature review.

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
