# Peer review of "Applications, challenges, and solutions of unmanned aerial vehicles in smart city using blockchain"

_PeerJ Computer Science, doi:10.7717/peerj-cs.1776_

## Round 0.1 · original submission · Major Revisions

Please follow review comments.

Reviewer 1 has suggested that you cite specific references. You are welcome to add it/them if you believe they are relevant. However, you are not required to include these citations, and if you do not include them, this will not influence my decision.

**Language Note:** The review process has identified that the English language must be improved. PeerJ can provide language editing services - please contact us at copyediting@peerj.com for pricing (be sure to provide your manuscript number and title). Alternatively, you should make your own arrangements to improve the language quality and provide details in your response letter. – PeerJ Staff

Reviewer 1 ·

Basic reporting

This study explores the advantages and challenges of using UAVs in smart city development, including blockchain solutions for improved security and privacy.

Major:

1. UAV application embedded with blockchain is a very popular topic. Highlight your novelty based on the limitation of existing works. The literature view for the existing survey that covered that topic is also missing.
2. Authors needs to discuss limited battery issue in the discussion. Moreover, adding AI to improve the performance of UAVs is highly recommended.
3. Also, the authors need to add connectivity in the discussion both in the presence and absence of network connectivity.
4. Authors need to discuss more on blockchain as it is one of the highlighted contributions. A discussion of filtering UAV data and the process of storing it in the blockchain including suitable consensus algorithms is highly required. Also, the authors need to discuss policy standards for adopting UAVs in smart cities.

Minor:

1. Corrections are required for grammatical errors.
2. The latest blockchain-based UAV papers are missing. Some are mentioned as follows.

-> "A Digital Twin-Based Drone-Assisted Secure Data Aggregation Scheme with Federated Learning in Artificial Intelligence of Things," in IEEE Network, vol. 37, no. 2, pp. 278-285, March/April 2023, doi: 10.1109/MNET.001.2200484.
-> "FBI: A Federated Learning-Based Blockchain-Embedded Data Accumulation Scheme Using Drones for Internet of Things," in IEEE Wireless Communications Letters, vol. 11, no. 5, pp. 972-976, May 2022, doi: 10.1109/LWC.2022.3151873.
-> "A Blockchain-Based Artificial Intelligence-Empowered Contagious Pandemic Situation Supervision Scheme Using Internet of Drone Things," in IEEE Wireless Communications, vol. 28, no. 4, pp. 166-173, August 2021, doi: 10.1109/MWC.001.2000429.
3. Add a table for related work that covers the summary of each work including their limitations.
4. Add a summary at the end of each section to increase readability.

Experimental design

All comments are provided above.

Validity of the findings

All comments are provided above.

Reviewer 2 ·

Basic reporting

See the detailed comments

Experimental design

See the detailed comments

Validity of the findings

See the detailed comments

Additional comments

This article looks at the benefits and drawbacks of deploying UAVs in the conception, development, and management of smart cities. This paper describes the importance and advantages of deploying UAVs in designing, developing, and maintaining smart cities. This paper overviews UAV uses types, applications, and challenges. The desired topic is currently under consideration but this work has several limitations being a review/survey paper. Some of them are as follows but not limited to the following:

1. The abstract of this article is badly written which must be revised.
2. In the abstract, the authors stated that “presented blockchain approaches for addressing the given problems for UAVs in smart research,”; however, there is no new approach presented in the article. Just an overview of the already existing approaches.
3. The key contribution of this review is not clear. The authors must highlight the key contribution throughout the article, as well highlight the gap.
4. Introduction section is poorly written, it must be revised.
5. There is no clear gap. Why we need to present this review if there is already existing literature dealing with such issues?
6. A review article should begin with a clear introduction that describes its objective and scope, and should include a comprehensive yet focused study of previous research on a specific issue.
7. It should comprise a well-organized literature review, synthesizing and summarizing major findings from relevant studies.
8. Most of the figures are badly organized, some of them gives no information. The text of the figures must be consistent with the paper body text. Also, some figures are taken from others work without giving proper reference, just copy pasted.
9. Critical analysis and evaluation of the included research's strengths and flaws are required, as is an integrated discussion of common themes and trends. The article should conclude with a summary of significant findings and recommended topics for future research.
10. I strictly suggest the authors to add a separate table which summarize the existing work and your contribution. Also, the table clearly illustrate the gap between the existing literature and your work.
11. The authors should look at some standard articles and understand how to write a review/suvey article? Ans why we should write a review article? What we should target in a review article? How we should summarize the gap? The recent challenges and prospects.
12. Most of the abbreviations are repeated or wrongly placed, such as Internet of Things, UAV, and so on. Every term must be defined for the first time and then use the short abbreviation throughout the article.
13. There are also too many grammatical mistakes and typos that must be corrected with proofreading.
14. Throughout the article, there is no fluency or connection among the sections, sub-sections, and paragraphs.
15. The technical depth of the article is not adequate too.
16. I’m wonder whether this article is a review article, or survey article, or an SLR? Because the aim/objective is not clear, especially section 3 look like an SLR but the remaining sections are different.
17. This work is just an overview of an already existing works. Overall, there is no novelty in this work, just a collection of discussion on the existing literature. I strictly recommend the authors to first have a look at some quality articles and understand the writing and aim of writing an article. It’s just the waste of time of either the authors, editors, as well as the reviewers.

Annotated reviews are not available for download in order to protect the identity of reviewers who chose to remain anonymous.

---

## Round 0.2 · accepted · Accept

I am writing to inform you that your manuscript - Applications, Challenges, and Solutions of Unmanned Aerial Vehicles in Smart City Using Blockchain - has been Accepted for publication. Congratulations!

Reviewer 1 ·

Basic reporting

I have no further concerns.

Experimental design

N/A

Validity of the findings

N/A